# Pre-training of U-Net Encoder for Improved Keypoint Detection in Transmitral Doppler Imaging

**Abas Abdi**[1]                                             abas.abdi@uwl.ac.uk
**Jevgeni Jevsikov**[1,2]                                    j.jevsikov@imperial.ac.uk
**Eman Alajrami**[1]                                         eman.alajrami@uwl.ac.uk
**Isreal Ufumaka**[1]                                        isreal.ufumaka@uwl.ac.uk
**Patricia Fernandes**[1]                                    Patricia.Fernandes@uwl.ac.uk
**Nasim DadashiSerej**[1]                                    nasim.dadashiserej@uwl.ac.uk
**Darrel P Francis**[2]                                      darrel@drfrancis.org
**Massoud Zolgharni**[1,2]                                   massoud@zolgharni.com

[1] *School of Computing and Engineering, University of West London, London, United Kingdom*
[2] *National Heart and Lung Institute, Imperial College, London, United Kingdom*

## Abstract

Self-supervised learning enables models to extract meaningful representations from unlabelled data. These representations can then be effectively transferred to supervised learning tasks, often requiring less labelled data compared to traditional approaches. The BT-Unet method leverages the strengths of U-Net for medical image segmentation tasks and is specifically designed to facilitate Barlow twins pre-training of backbone networks, such as ResNet50. However, accurate keypoint detection in medical images remains a challenge. This study investigates the potential of pre-training these backbones with unlabelled, domain-specific medical imagery using the Barlow Twins method to enhance keypoint detection performance in BT-Unet. We hypothesise that pre-training with domain-specific data will lead to more accurate and robust detection of Doppler peak velocities in Mitral Inflow ultrasound images compared to models trained without pre-training.

## 1. Introduction and Related works

Transfer learning, a staple in medical image analysis for scenarios with scarce labelled data, leverages models pre-trained on extensive datasets like ImageNet for fine-tuning on specific medical imaging tasks. Tajbakhsh et al. (Tajbakhsh et al., 2016) and Devan et al. (Devan et al., 2019) underscored its efficiency, highlighting superior performance compared to models trained from scratch, particularly in label-limited environments.

The distinct nature of medical versus natural images spurred the exploration of domain-specific pre-training. Ray et al. (Ray et al., 2022) showed that models pre-trained on medical datasets, such as histopathology images, outperform those pre-trained on generic datasets like ImageNet, in terms of both performance and convergence speed. This suggests that domain-specific pre-training could be more beneficial for medical imaging tasks, including keypoint detection. Complementing this, Krishnamoorthy, Agrawal, and Agarwal (Krishnamoorthy et al., 2024) explored self-supervised representation learning for diagnosing cardiac abnormalities from echocardiograms. Their findings align with our study, showing the potential of self-supervised learning to enhance diagnostic accuracy by effectively leveraging unlabelled data for feature extraction. This underscores the promise of domain-specific pre-training and self-supervised learning approaches in refining diagnostic precision and operational efficiency in healthcare.

## 2. Method

This study investigates the efficiency of initialising encoder backbones in the Unet architecture with the Barlow Twins Unet self-supervised learning (SSL) approach (Punn and Agarwal, 2022) to improve keypoint detection performance in Transmitral Doppler imaging. A 128-dimensional projection head, specifically designed for BT pre-training, is incorporated into the pre-trained neural network architectures, including ResNet50, MobileNetV2, and EfficientNetB0.

These architectures are pre-trained on a large, domain-specific dataset of 7,227 unlabelled Doppler mitral images. Subsequently, these pre-trained models undergo further fine-tuning using supervised learning on a smaller dataset of 1063 labelled Doppler mitral images. A detailed description of the dataset can be found in Jevsikov et al. (Jevsikov et al., 2024).

Early stopping is used during training of all models. An evaluation employing metrics like precision, recall, F1-score, and bias/standard deviation across E-wave and A-wave is conducted. Our results demonstrate a significant performance improvement, particularly with the unfrozen EfficientNetB0 model, compared to models trained solely with supervised learning. These findings highlight the potential of adapting pre-trained models with BT-Unet and SSL for specific medical imaging tasks.

The Barlow Twins objective function is defined as follows:

$$\mathcal{L}_{BT} = \sum_i (1 - C_{ii})^2 + \lambda \sum_i \sum_{j \neq i} C_{ij}^2 \tag{1}$$

where $C$ is the cross-correlation matrix between the outputs of twin networks. The first term penalizes deviations of diagonal elements from 1, ensuring feature integrity, while the second term, weighted by $\lambda$, minimizes off-diagonal elements to reduce feature redundancy.

## 3. Results

Table 1 showcases the comparative performance of U-Net encoder architectures—EfficientNetB0, MobileNetV2, and ResNet50—under various fine-tuning strategies, including supervised training and pre-training with the Barlow Twins method, followed by fine-tuning with both frozen and unfrozen weights.

The EfficientNetB0 model, with unfrozen weights, emerged as the standout performer, demonstrating superior precision (**0.91** E, **0.93** A), recall (**0.92** E, **0.81** A), and F1-scores (**0.92** E, **0.87** A), alongside the least bias (**-0.05**) and lowest standard deviation (**4.52**) for A-wave detection. This underscores the value of adaptive fine-tuning, where updating pre-trained backbone weights during supervised learning markedly enhances model accuracy and consistency.

Conversely, models with frozen backbones generally showed reduced effectiveness, highlighting the necessity for models to adapt during fine-tuning. Although EfficientNetB0 led in performance, ResNet50 and MobileNetV2 also delivered competitive results under various conditions. Notably, ResNet50 achieved an F1-score of **0.87** for E-wave with unfrozen weights, while MobileNetV2's best A-wave detection F1-score was **0.78** when frozen.

These insights confirm the crucial influence of backbone choice and fine-tuning approach on model performance in medical image analysis, with the unfrozen EfficientNetB0 model presenting a particularly promising option for high-precision tasks.

## 4. Conclusion

This study demonstrates the potential of Unet encoder pre-training using the Barlow Twins method to improve keypoint detection in Transmitral Doppler imaging. Notably, the unfrozen EfficientNetB0 model showcased exemplary performance metrics—achieving high precision, recall, and F1 scores, along with minimal bias and the lowest standard deviation for A-wave—underscoring the substantial benefits of fine-tuning pre-trained models for specific medical imaging tasks.

Our results emphasise the superiority of domain-specific pre-training, as opposed to more general approaches, providing a promising avenue to enhance diagnostic precision and operational efficiency through the use of unsupervised learning on unlabelled datasets.

Looking forward, further investigations could include experimenting with different projection head sizes, incorporating semi-supervised learning techniques to optimise training, and applying these pre-training methods to awider array of medical imaging types. These future research efforts hold the potential to broaden the scope of advanced diagnostic tools, significantly improving automated diagnostic capabilities and patient care within the healthcare sector.

Table 1: Precision, Recall and F1 Score for detection of peak velocity waves by the automated model, when compared to the human experts. For the correctly detected cases, levels of agreement between the model and the human experts was measured in terms of Bias (mean of differences) and Standard Deviation. Unfrozen: backbone weights are unfrozen during supervised fine-tuning; Frozen: backbone weights are frozen during supervised fine-tuning; Supervised: no pre-training applied.

| Model | Metrics (Precision/Recall/F1) | | Bias $\pm$ StD (cm/s) | |
|---|---|---|---|---|
|  | E | A | E | A |
| **EfficientNetB0** |  |  |  |  |
| Unfrozen | **.91/.92/.92** | **.93/.81/.87** | $2.65 \pm 8.36$ | **-0.05 $\pm$ 4.52** |
| Frozen | .71/.79/.75 | .82/.66/.73 | $2.01 \pm 11.14$ | $0.85 \pm 7.36$ |
| Supervised | .85/.90/.87 | .92/.78/.84 | **1.74 $\pm$ 5.17** | $0.56 \pm 8.15$ |
| **MobileNetV2** |  |  |  |  |
| Unfrozen | .79/**.81**/.80 | .84/.63/.72 | $1.83 \pm 8.83$ | -0.53 $\pm$ **4.47** |
| Frozen | .78/.79/.78 | .81/**.75/.78** | **1.03 $\pm$ 8.29** | **0.18 $\pm$** 13.37 |
| Supervised | **.92**/.79/**.85** | **.87**/.69/.77 | $4.54 \pm 12.57$ | $5.02 \pm 20.39$ |
| **ResNet50** |  |  |  |  |
| Unfrozen | .85/**.88/.87** | .89/.74/.81 | $1.89 \pm$ **5.18** | $0.54 \pm 12.33$ |
| Frozen | .71/.86/.78 | .81/.71/.75 | **1.45 $\pm$** 6.40 | **0.17 $\pm$** 11.34 |
| Supervised | **.88**/.87/**.87** | **.92/.77/.84** | $2.19 \pm 12.78$ | $0.25 \pm$ **5.25** |

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
