# OpenReview forum: "Pre-training of U-Net Encoder for Improved Keypoint Detection in Transmitral Doppler Imaging"
_MIDL.io/2024/Short_Papers — MIDL 2024 Short Papers_

### Official Review · Reviewer_pMPm · 2024-04-24

**Confidence:** 4
**Final Rating:** 5

**Review:**

In this paper, the authors investigate the potential of pre-training different backbones networks with unlabelled domain-specific medical images using the Barlow Twins method to enhance keypoint detection performance in BT-Unet. They used different pre-trained neural networks architectures (ResNet50, MobileNetV2 and EfficientNetB0)  on Dopplet mitral images.
Even if further experiments could be added to optimize training and to extend it to other medical imaging types, this paper shows the importance of the backbone choice and fine-tuning approach on model performance, and the promising results obtained using the unfrozen EfficientNetB0.

---

### Decision · Program_Chairs · 2024-04-26

Accept